# Rhizoboxes as Rapid Tools for the Study of Root Systems of *Prunus* Seedlings

**DOI:** 10.3390/plants11162081

**Published:** 2022-08-09

**Authors:** Ricardo A. Lesmes-Vesga, Liliana M. Cano, Mark A. Ritenour, Ali Sarkhosh, José X. Chaparro, Lorenzo Rossi

**Affiliations:** 1Horticultural Sciences Department, Indian River Research and Education Center, Institute of Food and Agricultural Sciences, University of Florida, Fort Pierce, FL 34945, USA; 2Plant Pathology Department, Indian River Research and Education Center, Institute of Food and Agricultural Sciences, University of Florida, Fort Pierce, FL 34945, USA; 3Horticultural Sciences Department, Institute of Food and Agricultural Sciences, University of Florida, Gainesville, FL 32603, USA

**Keywords:** peach, rootstock, rhizobox, root scanning, root system architecture, root angle

## Abstract

Rootstocks are fundamental for peach production, and their architectural root traits determine their performance. Root-system architecture (RSA) analysis is one of the key factors involved in rootstock selection. However, there are few RSA studies on *Prunus* spp., mostly due to the tedious and time-consuming labor of measuring below-ground roots. A root-phenotyping experiment was developed to analyze the RSA of seedlings from ‘Okinawa’ and ‘Guardian’™ peach rootstocks. The seedlings were established in rhizoboxes and their root systems scanned and architecturally analyzed. The root-system depth:width ratio (D:W) throughout the experiment, as well as the root morphological parameters, the depth rooting parameters, and the root angular spread were estimated. The ‘Okinawa’ exhibited greater root morphological traits, as well as the other parameters, confirming the relevance of the spatial disposition and growth pattern of the root system.

## 1. Introduction

The traits associated with root system architecture (RSA) can be employed as useful phenotyping tools for the breeding selection of plant materials, such as fruit-tree rootstocks. However, the study of underground root systems is more challenging than the study of the aerial part of the plant in several respects [1]. Soil is a complex medium that presents a more heterogeneous environment than the atmosphere. The variability of soil profiles in terms of physical, chemical, and biological conditions, as well as the pedological perturbation caused by planting and soil preparation, are variables to consider for RSA studies [2]. Currently, knowledge about root growth is relatively limited because of the complex dynamics of root systems compared to above-ground organs [3].

Rootstock selection plays a fundamental role in orchard management and successful fruit production. Some horticultural aspects to consider for rootstock breeding programs include soil adaptation, water and nutrient uptake, plant nutritional status, scion growth, pathogen resistance, and fruit quality [4,5]. ‘Guardian’™, a commercial peach *Prunus persica* rootstock, was released in 1993 through the cooperative efforts of the USDA-ARS and Clemson University [6]. This rootstock provides longer tree life to overcome the problem of peach-tree short life (PTSL) [7]. In the last decade, ‘Guardian’™ has been the predominant rootstock used for commercial peach production in the Southeastern United States, especially in South Carolina and Georgia. It is increasingly planted as a rootstock for almonds in California [8,9,10] with satisfactory field performance [11]. However, ‘Guardian’™ is not a recommended rootstock for peach production in Florida due to its susceptibility to *Meloidogyne floridensis*. On the other hand, ‘Okinawa’, a rootstock introduced in the United States in 1953 from Ryuku Islands (Japan), is one of the rootstocks utilized as genetic material in the University of Florida stone-fruit-breeding program for low-chill adaptation [12]. This low-chill rootstock is resistant to root-knot nematodes and exhibits good compatibility with several peach scion cultivars [13]. Despite these desirable features, ‘Okinawa’ rootstock is also not recommended for peach production in Florida because of its susceptibility to *M. floridensis* [14,15]. However, it is widely used outside of the United States, in countries such as Brazil [16] and Egypt [17].

Root architectural traits are fundamental for soil exploration, which influences the ability of plants to uptake below-ground resources [18]. Architectural traits are also strongly related to plant adaptation to sub-optimal conditions [19]. For instance, the most recognized contribution of RSA to water uptake is the ability to explore deeper soil layers. It is generally assumed that deeper root systems provide access to water stored deep in the soil [20]. Thus, deeper soil exploration contributes to drought resistance. Indeed, multiple studies provide evidence for the association between rooting depth and drought resistance in a panel of species, including trees [21]. However, the overall advantage of the root architectural traits must be interpreted in the context of the environment where the plants are established [19].

Among the geometric properties of root architectural traits, the root angle plays an important role in shaping the RSA of many species [19,22,23]. This is key, since it is one of the main traits contributing to rooting depth [24] and is strongly associated with the temporal and spatial acquisition efficiency of soil resources [25]. Moreover, the insertion angle of basal roots in dicotyledons determines the depth of root distribution; wider basal root angles determine shallower root systems [26,27]. For instance, a wider basal root angle in dicotyledons, such as the common bean (*Phaseolus vulgaris*), develops a shallower root system, enhancing topsoil foraging and phosphorous acquisition [19]. In contrast with monocotyledons such as rice (*Oryza sativa*), Kato et al. [28] demonstrated that the growth angle affects the vertical distribution and the rooting depth. A narrower angular spread indicates a deeper root system, which is advantageous for tolerance and adaptation to water-limited soil conditions [29]. Similarly, Manschadi et al. [19] demonstrated that in another monocotyledon, such as wheat (*Triticum aestivum*), drought-tolerant genotypes exhibit a narrower angular spread of seminal axes. Although the studies presented involved several species of economic importance, there is a lack of research evidence with regards to fruit-tree species, particularly rootstocks. Overall, the selection of the root angle may help to identify genotypes with a RSA more adapted to drought conditions, representing promising traits that can be exploited in crop-breeding programs [19].

Most plants repeat either their aerial or below-ground architectural units through reiteration, a morphogenetic process through which the organism duplicates its own elementary architecture entirely or partially [30]. Few attempts have been made to describe the genotypic variation in the root architecture of mature dicotyledon fruit-tree crops at the quantitative level; most of these have been on monocotyledon cereal crops [25]. Very few studies have been performed on *Prunus* spp., and in less detail. Through reiteration, the plant expresses its growth pattern repeatedly, from seedling to maturity [30]. Therefore, the analysis of the RSA at early developmental stages (seedlings) allows the measurement of root traits under controlled and relatively uniform conditions [31], especially in perennial trees such as *Prunus* spp. The quantification of root architectural traits becomes increasingly challenging as the plant grows. The root-insertion angles of seedlings may facilitate the use of root phenotyping as selection criteria in many cultivated species, since at early growth stages, this trait determines the development and performance of the mature root system. This has been demonstrated in the root angle of wheat genotypes, which developed either deeper or shallower root systems, depending on the root angle (narrower or wider) exhibited at the seedling growth stage [32,33].

Since most of the roots are below-ground and/or require a dark environment to grow naturally, root phenotyping requires special techniques. Rhizotrons (i.e., underground viewing chambers that offer a window of the root system) and, more recently, rhizoboxes (i.e., individual planting in thin translucent containers), are non-destructive techniques developed to perform direct and repeated measurements on the root system and high-quality imaging of the rhizosphere [34].

Rhizoboxes can present in many shapes and sizes, all utilizing translucent walls that allow the observation of the natural growth and development of an individual plant’s roots. Typically, these rhizoboxes are flat to ensure the visualization of the root architecture and to allow for them to be scanned or pictured. Of course, this greatly diminishes the growing space for the roots, and therefore cannot be used as a direct comparison to field conditions; however, we are able to observe RSA traits and root characteristics from which we can extrapolate to form a better understanding of future developments in the field, or to identify particular traits of interest in breeding. According to Mašková and Klimeš [35], flat rhizoboxes did not affect plant growth in terms of total biomass, nor did they affect root-system-growth comparisons. This form of direct monitoring allows the non-invasive quantification and estimation of root-trait development, since it does not disturb the spatial disposition of roots. Rhizoboxes combine the controlled conditions of the laboratory with field-oriented research, even when the soil environment is artificial [36]. The imaging methods used in rhizoboxes help to study the development of root systems, as well as rhizosphere dynamics, in different growth media. This way, in rhizoboxes, the RSA can be analyzed holistically, and promising traits can be applied to breeding [37]. The main objective of this study was to analyze the root systems of peach rootstock seedlings with morphological and geometric approaches using rhizoboxes. 

## 2. Results

### 2.1. Whole-Root System Depth: Width Ratio

Overall, the ‘Guardian’™ whole-root system was significantly deeper (Figure 1a) and wider (Figure 1b) than the ‘Okinawa’ during the study. The depth:width ratio (D:W) of the ‘Okinawa’ was higher than that of the ‘Guardian’™ each week for the duration of the experiment (Figure 1c); however, there were no significant differences throughout the 7-week period.

### 2.2. Root-Growth Parameters

The seedlings of the ‘Okinawa’ had significantly longer roots (Figure 2a) and a greater total-root-surface area (Figure 2b), root volume (Figure 2d), number of root tips (Figure 2e), and number of root forks (Figure 2f), compared with the ‘Guardian’™ (789 cm). However, the average root diameter of the ‘Okinawa’ was significantly lower than that of the ‘Guardian’™ (Figure 2c). The significant linear regressions between root-growth parameters in the two different cultivars are reported in the Appendix A.

### 2.3. Total-Root-Length Distribution by Diameter Classes

The total root length was divided into three diameter classes: very fine (≤0.5 mm), fine (>0.5 to ≤1.0 mm), and large (>0.5 to ≤1.0 mm). The very fine root length of the ‘Okinawa’ (1955.5 cm) was significantly greater than that of the ‘Guardian’™ (611.1 cm). There were no significant differences in root length for the fine and large root diameters between the cultivars. (Table 1).

### 2.4. Root-Depth Pattern and Root-Depth Index

Root horizon A of the ‘Okinawa’ represented a significantly larger portion of the total root length (44.2%) compared to horizons C (14.8%) and D (15.1%), but was not significantly different from horizon B (25.9%). Furthermore, there were no significant differences in root length between horizons B, C, and D (Figure 3a). Similarly, root horizon A of the ‘Guardian’™ presented a root-length percentage (63.36%) that was significantly higher than that of horizons B (28.4%), C (9.8%), and D (7.12%). Moreover, horizon B was significantly higher than horizons C and D in this cultivar for this parameter, whereas horizons C and D were not significantly different (Figure 3b). Finally, the root-depth index (RDI) of the ‘Okinawa’ (15.09) was significantly higher than that of the ‘Guardian’™ (9.58) (Figure 3c).

### 2.5. Root Spreading Angle

The total root length was not significantly different in the shallower (0–25°), shallow (25–45°), or deeper spreading angles (65–90°). However, the total root length was significantly different in the deep (45–65°) spreading angle between the ‘Okinawa’ (303.35 cm) and the ‘Guardian’™ (168.08 cm). On the other hand, each cultivar presented the same pattern between spreading-angle ranges; the shallower and shallow spreading angles were not significantly different from each other, but significant differences were observed for the deep and deeper spreading angles (Table 2).

## 3. Discussion

There were significant statistical differences (*p* ≤ 0.05) between the rootstock cultivars for most of the RSA parameters analyzed. The estimation of the whole-root-system depth:width ratio (D:W) was proposed, from an allometric perspective, to assess the evolution of the proportion of depth vs. width, as a useful indicator of the root-system growth and architecture for future studies. The average depth and width reached by the whole-root system measured weekly in the ‘Guardian’™ seedlings were significantly higher than in the ‘Okinawa’. Conversely, the D:W showed no significant differences between cultivars or weeks. Moreover, the D:W decreased as the plants grew, with the ratio in the ‘Okinawa’ higher than that in the ‘Guardian’™, with significant differences. Furthermore, the D:W of the ‘Okinawa’ tended to be more stable than that of the ‘Guardian’™, with the D:W of the ‘Guardian’™ decreasing sharply earlier (week 4) than in the ‘Okinawa’ (week 7).

Significant differences were also observed between the root-growth parameters of both rootstock cultivars. Except for the average root diameter, all the root-growth parameters of the ‘Okinawa’ seedlings were significantly higher than those of the ‘Guardian’™. The linear regressions performed (reported in Appendix A) showed similar significant patterns except for the root diameter. These results contrasted with the average of the whole-root-system depth and width reached by the ‘Okinawa’ seedlings, which were significantly lower than those of the ‘Guardian’™. Therefore, it can be inferred that the depth and width reached by the root system alone is a poor descriptor of the actual root-system growth.

Despite both cultivars belonging to the same species, *P. persica*, these results confirm the contrast that can be found between rootstock cultivars. It is noteworthy that no previous studies on the root-system architecture (RSA) of the ‘Okinawa’ rootstock nor the ‘Guardian’™ rootstock have been published. Although there are reports that mention the ‘Guardian’™ as a rootstock that imparts excellent scion vigor and productivity [38,39], they do not describe its RSA. Similarly, this also occurs with the ‘Okinawa’ rootstock. The RSA features of these rootstocks confirm the findings from parallel studies that mention the ‘Okinawa’ as a rootstock that transmits high vigor to the scion cultivar [16]. In Northern Thailand, it was reported that ‘Okinawa’ induced good scion performance and reached the highest tree growth compared to other rootstocks, such as ‘Khunwang’, ‘White Angkhang’, ‘Red Angkhang’, and ‘Flordaguard’ [40]. This is consistent with the study by Kucukyumuk and Erdal [41], which indicates that vigorous fruit rootstocks have large root systems. 

The morphology of the root system is as important as the actual soil volume explored by the roots [42]. The significantly greater length of the very fine roots (≤0.5 mm) of the ‘Okinawa’ is a significant morphological trait, since knowledge of root diameters can provide important information about the root penetration and exploration potential in relation to the soil-pore size. Fine roots (diameter < 1 or 2 mm) are believed to play an important role in water and mineral uptake in higher plants [43,44]. However, it is necessary to keep in mind that this does not guarantee the absorption ability of the root system, since very fine roots may be suberized or even lignified [45].

In addition to the growth and morphological parameters, the spatial arrangement and distribution of the roots is another key aspect of the RSA. The D:W ratio estimation of the root systems studied in this chapter revealed little information and few contrasts between the two rootstocks. Nevertheless, these results agreed with the root-distribution patterns (RDPs) of both cultivars; the ‘Okinawa’ exhibited a more deeply distributed root system than the ‘Guardian’™. Most of the total root lengths of the ‘Guardian’™ seedlings were concentrated in the upper level, whereas in the ‘Okinawa’ seedlings, they were more deeply distributed. In addition, the percentage of the root length distributed at deeper horizons was almost twice as high in the ‘Okinawa’ than in the ‘Guardian’™, where more than half of the total root length was in the top horizon (A). These results were confirmed by the superior root depth index (RDI) of the ‘Okinawa’, reflecting the deeper root system of this cultivar. The differences in root spreading angle (RSG) between the cultivars indicate that the root system of the ‘Okinawa’ tends to grow not only deeper due to its RDP and RDI, but also downwards. This trait is key for soil exploration in the pursuit of resources [25,32]. Solari et al. [46] confirmed that tree vigor is positively related to rootstock hydraulic properties. Moreover, such RSA traits can explain the differences in nutrient concentrations among rootstocks. In general, the root system architecture is different in *Prunus* from one rootstock cultivar to another, influencing their nutrient-uptake ability [41], which is evidenced by the nutritional status of the scion [47,48]. Rootstocks have a significant influence on leaf-mineral composition. Thus, according to Kumar et al. [49], the higher nutrient concentration in scion leaves is induced by invigorating rootstocks. The same authors reported similar findings in citrus rootstocks, indicating a significant positive correlation between the nitrogen (N), Ca, K, and Mg concentrations in leaves with the total root length, surface area, root volume, and number of tips. This was found by Shahkoomahally et al. [4] in the ‘UFSun’ peach on ‘Okinawa’, which showed greater concentrations of calcium, potassium, and magnesium in leaves than scions from the same cultivar grafted on other rootstocks, inducing a tree nutritional balance that was superior to those of other commercial rootstocks, maintaining the highest concentrations of macronutrients throughout the year. Given its better mineral-uptake efficiency, this study suggests the use of ‘Okinawa’ for sandy soils. Our results are comparable with those in the study by Ahmed et al. [17], where scions grafted on ‘Okinawa’ in sandy soils achieved superior percentages of initial and final fruit set. Similarly, it has been reported that ‘Chimarrita’ peach on ‘Okinawa’ reached higher levels of Brix, L-ascorbic acid, and carotenoids [50] than other rootstocks. Furthermore, the ‘Okinawa Clone 1’ rootstock induced greater Brix levels in ‘Maciel’ peach [51].

Since this study was performed on young plants, it can be considered too early to infer the potential performance of these rootstocks (‘Okinawa’ and ‘Guardian’™); however, it is possible that the plant express the growth pattern shown in its seedling stage in maturity through reiteration [30]. This reiteration ability has been observed in oil-palm (*Elaeis guineensis* Jacq.) root systems [52]. It has been found that reiteration occurs in the entire root system of other stone fruit species, such as plum (*Prunus cerasifera*) [53], implying stone fruit species are not exceptions to this trait.

In summary, a larger root system may influence fruit quality beyond scion nutritional status. The measurement of the D:W of the root system alone is not a good indicator of the actual root-system size; however, the ratio can suggest how the root system grows, which is as relevant as the root-system morphology.

## 4. Materials and Methods

### 4.1. Plant Material, Seeds Stratification and Germination

Seedlings of two peach rootstock cultivars of *P. persica* used in commercial peach production in the United States (‘Okinawa’ and ‘Guardian’™) were propagated from seeds provided by the Fruit Tree Breeding and Genetics Laboratory of the University of Florida in Gainesville, FL, USA. ‘Okinawa’ is a rootstock cultivar selected from landrace seed imported from Ryuku, Japan, and is also highly homozygous. ‘Guardian’™ is a seed-propagated rootstock that is highly homozygous and uniform in performance.

The seeds were submerged in water, and those that floated were discarded. The remaining seeds were left in water for 96 h for stratification, and the water was renewed every 24 h. After 96 h, the seeds were submerged in a solution of water with Captan fungicide at 0.15% (*w/v*) for another 24 h. After this period, the solution was drained and the seeds were stored in a plastic bag with perlite, previously moistened with the same fungicide solution. The seed bags were stored at 4–8 °C until germination for a period of 6 weeks.

After germination, the seeds were sown in plastic trays (720700C SureRoots^®^; T.O. Plastics, Inc.; Clearwater, MN, USA) that consisted of star-shaped deep cell plugs (12.7 cm depth × 5.1 cm top width) containing a mixture (1:1) of potting-mix sphagnum (Jolly Gardener^®^ Pro-Line C/20 Growing Mix; Jolly Gardener Products, Inc.; Poland Spring, ME, USA) and coarse perlite (Specialty Vermiculite Corp.; Pompano Beach, FL, USA). The substrate mixture was previously autoclaved at 121 °C for 90 min. The germination trays were maintained within a plastic-covered greenhouse located in Fort Pierce, FL, USA, at 27°25′34.2′′ N–80°24′34.0′′ W, covered with a shade cloth at 70% of shading, and watered manually. The seedlings were grown under greenhouse conditions. The average temperature was 20–30 °C during the day and 15–25 °C at night. Relative humidity (RH) was an average of 70–80% and photoperiod followed a natural phase of roughly 12–12 h (day–night).

### 4.2. Seedling Growing Conditions

After 10 days, at the emergence stage, the seeds that exhibited a 3–5-centimeter-long healthy and straight radicle were selected and sown in rhizoboxes (40 cm × 40 cm × 2.5 cm) (Figure 4a). One seed was transplanted per rhizobox. The rhizoboxes consisted of two clear glass panes (40 cm × 40 cm) attached to both faces of a three-sided plastic frame with a perforated bottom to allow water filtration. The plates and plastic frames were previously disinfected with a solution of sodium hypochlorite (1.5% *v/v*) in water for 30 min. The rhizoboxes were filled with a mixture of potting-mix sphagnum:coarse perlite (3:1) from the same manufacturers as those used for germination trays. Furthermore, the substrate was blended with the controlled-release (3 months) fertilizer, Osmocote^®^ Plus (15-9-12) (A.M. Leonard, Inc., Piqua, OH, USA), at 0.5% (*v/v*).

The emergent seeds were transplanted next to one of the glass panes of the corresponding rhizobox. Both panes of each rhizobox were covered with aluminum foil to exclude light and keep the roots under dark conditions. The rhizoboxes with the seeds were set in black plastic containers with bottom-draining holes, standing on wooden racks reclined at 30°. The experiment was maintained under laboratory conditions, with 23 °C and 65% relative humidity (RH. Complementary light-emitting diode (LED) bulbs and high-pressure sodium (HPS) lamps were adapted to keep the photoperiod at 16/8 h (day/night). The seedlings were watered manually throughout the experiment daily.

### 4.3. Root Systems Scanning

The root systems were scanned from the rhizoboxes once a week for 7 weeks with a flatbed scanner EPSON Expression 10000XL (EPSON America, Inc., Los Alamitos, CA, USA). Full color images were captured in Tagged Image Format File (TIFF) at resolution of 400 dots per inch (dpi). After this period, the root systems of the seedlings were extracted for measurement by removing one of the glass panes from the corresponding rhizobox. A pinboard was used to remove each plantlet, maintaining the spatial distribution of the roots. The pinboard consisted of an acrylic panel with 2.5-centimeter-long nails separated by 2.5 cm × 2.5 cm. Another acrylic panel with holes that corresponded with the nails was matched as a grid on the pinboard. The pinboard-grid was inserted into the rhizoboxes at the sampling moment (Figure 4b). The substrate was washed off the roots inserted in the pinboard by gently spraying with water.

After the root systems were extracted from the rhizoboxes, these were split into 4 horizons (A, B, C, and D, from top to bottom) of 10 cm (Figure 4a). The roots from each horizon were scanned with a flatbed scanner EPSON Perfection V800/V850 (EPSON America, Inc., USA) by placing them in a Plexiglas tray (20 cm × 30 cm × 1.5 cm). The plexiglass tray contained water to untangle the roots and minimize their overlapping. The images were captured in TIFF at 600 dpi resolution. Subsequently, the central portion of the top horizon (A) within an area of 10 cm × 10 cm was scanned with the EPSON Perfection V800/V850.

### 4.4. Image-Analysis Software and Measurements

From the images obtained every week with the EPSON Expression 10000XL scanner, the depth (D) and width (W) reached by the whole-root system were measured, and the depth:width ratio (D:W) was estimated. The images obtained with the EPSON Perfection V800/V850 scanner were analyzed using the root-image-analysis software, WinRHIZO™ Pro (Regent Instruments Inc., Quebec City, QC, Canada) (Figure 4c). The measurements of the root-growth parameters were total root length (cm), total root-surface area (cm^2^), average root diameter (mm), total root volume (cm^3^), number of root tips, and number of root forks. Additionally, the root length was estimated for each root diameter class, following the criteria applied by Caruso et al. [54]: very fine (≤0.5 mm), fine (from >0.5 mm to <1.0 mm), and large (>1.0 mm).

Based on the total root length from each horizon, the root-distribution pattern (RDP) and the root-depth index (RDI) were calculated, following the protocol proposed by Oyanagi et al. [33,55]. The RDP was estimated by calculating the percentages of the root length of each 10-centimeter horizon from the whole-root system. Subsequently, the percentage values were multiplied by the value of the middle depth (cm) of the corresponding horizon (5 for A, 15 for B, 25 for C, and 35 for D) and divided by 100.

To study the root spreading angles of each genotype, a slightly modified version of the protocol followed by Ramalingam et al. [56] was applied. The scanned images from the central portion of horizon A were split into four angular sections projected from the base of the seedling: shallower (0–25°), shallow (25–45°), deep (45–65°), and deeper (65–90°) (Figure 4d). Subsequently, the root length (cm) was measured from each angular section. 

### 4.5. Experimental Design and Statistical Analysis

The experiment was arranged in a completely randomized design with two cultivar treatments and sixteen single-tree replications per genotype. Each tree was an experimental unit. The data collected from the described RAS parameters were compared among cultivars by analysis of variance (ANOVA). The data were processed using RStudio software (2019), and a Tukey Honest Significance Difference (HSD) test was used to compare the means when the differences between treatments were significant (*p* ≤ 0.05).

## 5. Conclusions

Studying the architectural root traits of rootstocks provides higher detail on tree growth and development. To date, most peach rootstocks have been studied only for their field performance and productivity, which offers the industry a fundamental but short-term outlook. However, very little attention has been given to the RSA traits that can benefit the long-term breeding selection processes for stone-fruit rootstocks. The use of rhizoboxes allows the study of these root traits in controlled environments. The experiment conducted showed the potential of the use of rhizoboxes as a rapid tool for the study of the root systems of *Prunus* seedlings. However, field comparisons between the data obtained from rhizotrons, destructive analysis, and rhizoboxes are needed to further support the use of rhizoboxes as reliable tools in fruit-tree-species RSA studies.

## Figures and Tables

**Figure 1 plants-11-02081-f001:**
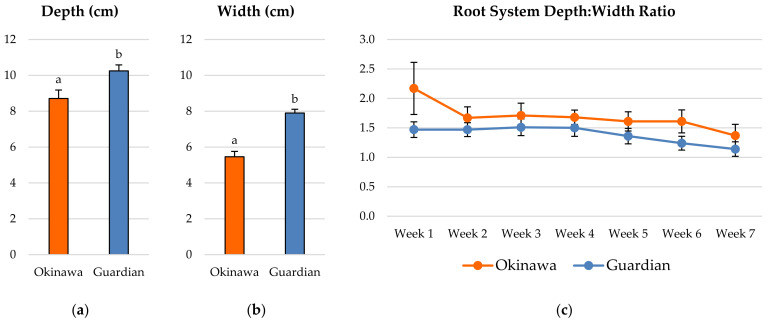
Average (**a**) depth and (**b**) width reached by the whole-root system of the ‘Okinawa’ and ‘Guardian’™. (**c**) Depth:width ratio of the whole-root system of the ‘Okinawa’ and ‘Guardian’™. Bars with different letters indicate significantly different values according to Tukey’s (HSD) test (*p* ≤ 0.05).

**Figure 2 plants-11-02081-f002:**
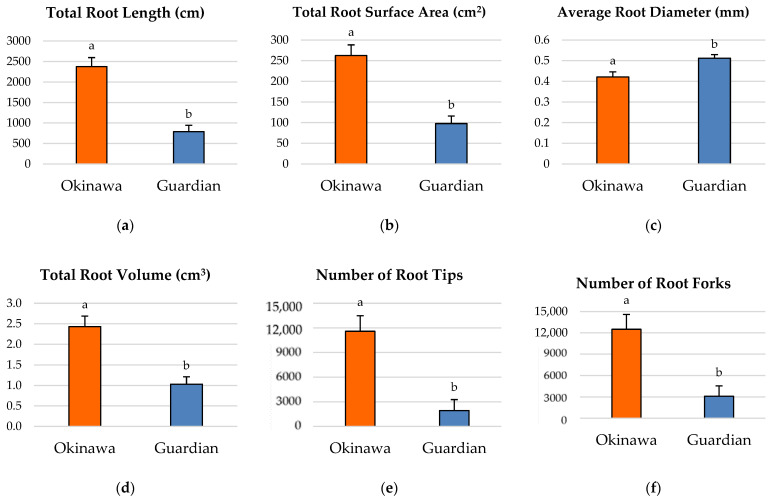
Comparisons of root morphological parameters in ‘Okinawa’ and ‘Guardian’™ rootstocks for (**a**) total root length, (**b**) total-root-surface area, (**c**) average root diameter, (**d**) total root volume, (**e**) number of root tips, and (**f**) number of root forks. Bars with different letters indicate significantly different values according to Tukey’s (HSD) test (*p* ≤ 0.05).

**Figure 3 plants-11-02081-f003:**
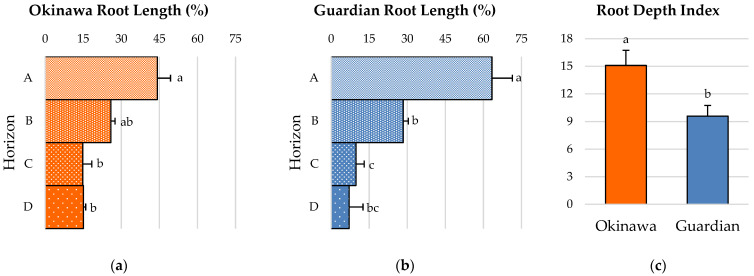
(**a**) Root-distribution pattern (RDP) in terms of root-length percentage of (**a**) ‘Okinawa’ and (**b**) ‘Guardian’™ rootstocks. (**c**) Root-depth index (RDI) of ‘Okinawa’ and ‘Guardian’™ rootstocks. Bars with different letters indicate significantly different values according to Tukey’s (HSD) test (*p* ≤ 0.05).

**Figure 4 plants-11-02081-f004:**
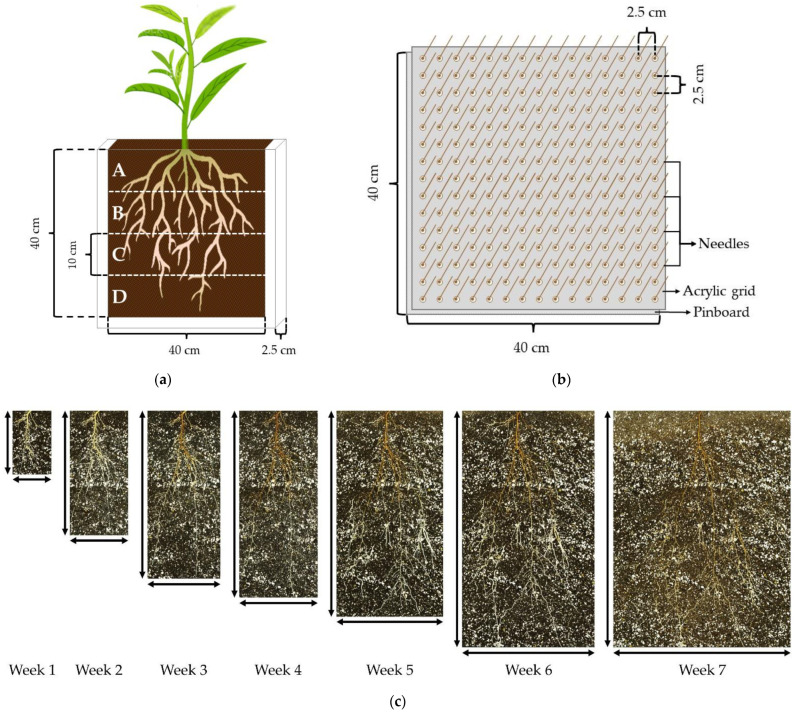
(**a**) Scheme of the rhizoboxes used to observe the seedling root system of ‘Okinawa’ and ‘Guardian’™ rootstocks (1 seedling/rhizobox). The dashed white lines indicate the four horizons (A, B, C, and D) into which the root systems were split for scanning with their respective dimensions. (**b**) Diagram of the pinboard used for root-system extractions from the rhizoboxes. The acrylic grid was fitted with the needles of the pinboard. (**c**) Images captured once a week for 7 weeks from a rhizobox with the root system of one seedling. The maximum depth (↕) and width (↔) reached by the whole-root system, from which whole-root system D:W was estimated. (**d**) Representation of the central portion of the top root horizon (A) from which the root spreading angle (RSG) was estimated. RSG was estimated by measuring the root length (cm) within each angular section: shallower (0–25°), shallow (25–45°), deep (45–65°), and deeper (65–90°).

**Table 1 plants-11-02081-t001:** Statistical analysis results of total root length (cm) distributed by root-diameter classes. The root-diameter classes are very fine (≤0.5 mm), fine (>0.5 to ≤1.0 mm), and large (>1.0 mm). Values not connected by the same letter are significantly different according to Tukey’s (HSD) test (*p* ≤ 0.05).

Root Length (cm) Distribution by Diameter Classes ^a^				
Diameter Class	Cultivar	Response	SE	Group
Very Fine (≤0.5 mm)	‘Okinawa’	1955.5	264.50	*a*
	‘Guardian’™	611.1	74.69	*b*
Fine (>0.5 to ≤1.0 mm)	‘Okinawa’	366.5	76.10	*bc*
	‘Guardian’™	154.7	27.15	*c*
Large (>1.0 mm)	‘Okinawa’	53.7	7.01	*c*
	‘Guardian’™	23.1	4.88	*c*

^a^ where SE = standard error.

**Table 2 plants-11-02081-t002:** Statistical analysis results of the root spreading angle from the central portion of the top horizon (A) of ‘Okinawa’ and ‘Guardian’™ rootstocks. The RSG was estimated in terms of the total root length distributed between angular sections: shallower (0–25°), shallow (25–45°), deep (45–65°), and deeper (65–90°). Values not connected by the same letter are significantly different according to Tukey’s (HSD) test (*p* ≤ 0.05).

Total Root Length (cm) by Spreading Angle from the Cultivar–Angle Interaction ^a^				
Angle	Cultivar	Response	SE	Group
Shallower (0–25°)	‘Okinawa’	17.89	4.51	*a*
	‘Guardian’™	7.15	3.88	*a*
Shallow (25–45°)	‘Okinawa’	103.57	16.84	*ab*
	‘Guardian’™	36.49	12.30	*a*
Deep (45–65°)	‘Okinawa’	303.35	42.92	*c*
	‘Guardian’™	168.08	22.31	*b*
Deeper (65–90°)	‘Okinawa’	460.17	29.89	*d*
	‘Guardian’™	382.70	22.72	*cd*

^a^ where SE = standard error.

## Data Availability

The data presented in this study are available within this article. Raw data are available upon request from the corresponding author (l.rossi@ufl.edu).

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
