# Peer review of "Rhizoboxes as Rapid Tools for the Study of Root Systems of Prunus Seedlings"

_plants, 2022, doi:10.3390/plants11162081_

Round 1
Reviewer 1 Report
The authors analyse the root system of the peach rootstocks 'Okinawa' and 'Guardian', using a system of 'rhizoboxes'. For this purpose, they study seedlings at different stages of development (1 to 7 weeks). Each week they analyse the root depth and root width of each seedling. At the end of this period, the authors carried out a more detailed study of the root system. The analysis of the morphological characteristics of the roots and the relationship between the depth and width of the root system indicate, according to the authors, that there are differences between the two rootstocks.
It is very important that the authors indicate whether the rootstocks are homozygous lines or not, as the seedlings they are analysing come from seeds and if the rootstocks are heterozygous, each seed may have a different genotype, including genes related to root system architecture. Therefore, each seedling could have a different genotype and show different phenotypes which, in my opinion, would invalidate the conclusions drawn.
On the other hand, the study of the RSA in the seedling stage allows the analysis of different parameters of the seedling in a precise and controlled manner. However, it is necessary to check that there is a good correlation with the RSA phenotype in the adult plant. Authors should provide evidence, obtained by themselves or others, showing the relationship between RSA at seedling and adult plant stage, so that their results can be extrapolated to the performance of rootstocks when used in the field.
Figure 1c does not show the roots well, try to improve it.
In previous studies on other species, significant correlations between RSA variables have been observed. It is suggested that the authors conduct a study of the correlations between the root system parameters they have analysed, e.g. between: total root length, root surface area, mean root diameter and total root volume.
Lines 140- 142. Please check this sentence: “whereas there were no significant differences between 141 these last two root diameter classes”, because the root length of the fine class in Okinawa is 366.5 and that of the large class is 53.7, i.e. about 7 times greater.
I am surprised that the SE values in several of the tables in the paper have the same values. Please, check the SE values in:
Table 1: Measurements of Root System Depth:Width Ratio for two peach culivars.
Table 3: Root Length (cm) Distribution by Diameter Classes
Table 4: ‘Okinawa’ Root Distribution Pattern (RDP). ‘Guardian’TM Root Distribution Pattern (RDP)
Table 5: Total Root Length (cm) by Spreading Angles between Angles. Total Root Length (cm) by Spreading Angles from the interaction Cultivar:Angle
Please, explain the different degrees of freedom indicated in the Tables and why they are different. It is not clear how many seedlings of each rootstock have been examined. Are the same seedlings examined throughout the 7 weeks and at the end of this period?
Author Response
Replies to Reviewer #1
The authors analyze the root system of the peach rootstocks 'Okinawa' and 'Guardian', using a system of 'rhizoboxes'. For this purpose, they study seedlings at different stages of development (1 to 7 weeks). Each week they analyze the root depth and root width of each seedling. At the end of this period, the authors carried out a more detailed study of the root system. The analysis of the morphological characteristics of the roots and the relationship between the depth and width of the root system indicate, according to the authors, that there are differences between the two rootstocks.
Thanks for taking the time to read our manuscript. Please find our replies to your comments below.
It is very important that the authors indicate whether the rootstocks are homozygous lines or not, as the seedlings they are analyzing come from seeds and if the rootstocks are heterozygous, each seed may have a different genotype, including genes related to root system architecture. Therefore, each seedling could have a different genotype and show different phenotypes which, in my opinion, would invalidate the conclusions drawn.
Information about the rootstocks homozygous characteristics have been added to the materials and methods section.
On the other hand, the study of the RSA in the seedling stage allows the analysis of different parameters of the seedling in a precise and controlled manner. However, it is necessary to check that there is a good correlation with the RSA phenotype in the adult plant. Authors should provide evidence, obtained by themselves or others, showing the relationship between RSA at seedling and adult plant stage, so that their results can be extrapolated to the performance of rootstocks when used in the field.
A paragraph has been added to the discussion, showing that reiteration usually provide good correlation with the RSA phenotype in the adult plan
Figure 1c does not show the roots well, try to improve it.
Figure 1c has been updated to a HD picture.
In previous studies on other species, significant correlations between RSA variables have been observed. It is suggested that the authors conduct a study of the correlations between the root system parameters they have analyzed, e.g. between: total root length, root surface area, mean root diameter and total root volume.
Significant correlations have been added and are available in the Supplemental Materials section.
Lines 140- 142. Please check this sentence: “whereas there were no significant differences between 141 these last two root diameter classes”, because the root length of the fine class in Okinawa is 366.5 and that of the large class is 53.7, i.e. about 7 times greater.
Sentence has been checked and rectified and new graphs and table have been provided.
I am surprised that the SE values in several of the tables in the paper have the same values. Please, check the SE values in:
SE errors values have been checked and rectified
Table 1: Measurements of Root System Depth:Width Ratio for two peach cultivars.
Table 3: Root Length (cm) Distribution by Diameter Classes
Table 4: ‘Okinawa’ Root Distribution Pattern (RDP). ‘Guardian’TM Root Distribution Pattern (RDP)
Table 5: Total Root Length (cm) by Spreading Angles between Angles. Total Root Length (cm) by Spreading Angles from the interaction Cultivar:Angle
Please, explain the different degrees of freedom indicated in the Tables and why they are different. It is not clear how many seedlings of each rootstock have been examined. Are the same seedlings examined throughout the 7 weeks and at the end of this period?
Following reviewers #3 comments the tables have been converted into graphs and some statistical inaccuracies have been fixed. All the data have now been double checked for statistical accuracy. Sixteen single-tree replications per genotype (n = 8) were used in the study. Although, the same seedlings were examined throughout the 7 weeks and at the end of the experiment, some of the Okinawa seedlings died and were only used for time points in which they were alive, this explains the differences in the degrees of freedom.
Reviewer 2 Report
The paper is fascinating and deals with interesting issues. The material and method are well-conceived. The results are presented in a clear and straightforward way. They are well discussed in relation to the results obtained in similar papers. The conclusions are clear and unambiguous.
The paper will be very interesting for the wider scientific public because it methodically covers issues from different scientific fields.
Author Response
Thanks for your kind words. We really appreciated the positive review.
Reviewer 3 Report
The manuscript “Rhizoboxes as Rapid Tools for the Study of Root Systems of Prunus Seedlings ” seems to be an excellent topic and time-worthy work indeed. The manuscript is well written and coherently presented, there are no methodological errors. I have only several comments and suggestions.
Introduction: The rhizoboxes are mentioned at the end of the introduction, I would suggest a more detailed description of the principle or current application of the system.
Line 268: Please describe the greenhouse conditions (such as temperature, humidity etc.).
Line 317-318: fine (> 0.5 ‒ ≤ 1.0 mm). Please change to fine (> 0.5mm, ≤ 1.0 mm).
Table 1-5: Can the existing data table express the experimental results intuitively? I suggest representing related data in the form of bar graphs using different colour. Or change the existing table format and display the analysis data of each indicator in the form of a separate table.
Author Response
Replies to Reviewer #3
The manuscript “Rhizoboxes as Rapid Tools for the Study of Root Systems of Prunus Seedlings ” seems to be an excellent topic and time-worthy work indeed. The manuscript is well written and coherently presented, there are no methodological errors. I have only several comments and suggestions.
Thanks for your kind words and for taking the time off your schedule to review our manuscript. Please find our detailed reply below.
Introduction: The rhizoboxes are mentioned at the end of the introduction, I would suggest a more detailed description of the principle or current application of the system.
A paragraph about rhizoboxes has been added.
Line 268: Please describe the greenhouse conditions (such as temperature, humidity etc.).
Greenhouse conditions have been added.
Line 317-318: fine (> 0.5 ‒ ≤ 1.0 mm). Please change to fine (> 0.5mm, ≤ 1.0 mm).
It has been changed to “> 0.5 mm to ≤ 1.0 mm”
Table 1-5: Can the existing data table express the experimental results intuitively? I suggest representing related data in the form of bar graphs using different color. Or change the existing table format and display the analysis data of each indicator in the form of a separate table.
The tables have been converted to graphs
Round 2
Reviewer 1 Report
The authors have modified the manuscript according to the recommendations given.
The replacement of some the tables by graphs, facilitates the visualisation of the results.
The origin of the plant material has also been clarified in the Materials and Methods section. In this new version it has been clarified that the two rootstocks are homozygous, otherwise the results would not be valid.
Errors and statistical deficiencies have also been corrected.
The main deficiency I find in the work is the number of seedlings analysed, which, in my opinion, is low.
Please, include in the Discussion section a remark on the significance of the linear regressions performed.
Author Response
The authors have modified the manuscript according to the recommendations given. The replacement of some the tables by graphs, facilitates the visualisation of the results. The origin of the plant material has also been clarified in the Materials and Methods section. In this new version it has been clarified that the two rootstocks are homozygous, otherwise the results would not be valid. Errors and statistical deficiencies have also been corrected.
Thank you for your time. Detailed answers to your comments can be found below.
The main deficiency I find in the work is the number of seedlings analysed, which, in my opinion, is low. Please, include in the Discussion section a remark on the significance of the linear regressions performed.
We understand that the number of seedling analyzed is a little bit low and that's why we decided to submit this manuscript for a Technical Note and not as a Research Paper. A remark in the Discussion section on the significance of the liner regression we performed is now added to the manuscript.